# Computer-aided discovery of a metal–organic framework with superior oxygen uptake

Peyman Z. Moghadam[1], Timur Islamoglu[2], Subhadip Goswami[2], Jason Exley[3], Marcus Fantham[1], Clemens F. Kaminski [1], Randall Q. Snurr[4], Omar K. Farha[2,4,5] & David Fairen-Jimenez [1]

Current advances in materials science have resulted in the rapid emergence of thousands of functional adsorbent materials in recent years. This clearly creates multiple opportunities for their potential application, but it also creates the following challenge: how does one identify the most promising structures, among the thousands of possibilities, for a particular application? Here, we present a case of computer-aided material discovery, in which we complete the full cycle from computational screening of metal–organic framework materials for oxygen storage, to identification, synthesis and measurement of oxygen adsorption in the top-ranked structure. We introduce an interactive visualization concept to analyze over 1000 unique structure–property plots in five dimensions and delimit the relationships between structural properties and oxygen adsorption performance at different pressures for 2932 already-synthesized structures. We also report a world-record holding material for oxygen storage, UMCM-152, which delivers 22.5% more oxygen than the best known material to date, to the best of our knowledge.

[1] Department of Chemical Engineering & Biotechnology, University of Cambridge, Philippa Fawcett Drive, Cambridge CB3 0AS, UK. [2] Department of Chemistry, Northwestern University, Evanston, IL 60208, USA. [3] Particulate Systems, Micromeritics Instrument Corp. 4356 Communications Drive, Norcross, GA 30093, USA. [4] Department of Chemical and Biological Engineering, Northwestern University, Evanston, IL 60208, USA. [5] Department of Chemistry, Faculty of Science King Abdulaziz University, Jeddah 21589, Saudi Arabia. Correspondence and requests for materials should be addressed to P.Z.M. (email: peymanzmoghadam@gmail.com) or to O.K.F. (email: o-farha@northwestern.edu) or to D.F-J. (email: df334@cam.ac.uk)

n the past decade metal–organic frameworks (MOFs)[1–4] have emerged as among the most promising materials for a wide range of applications including gas storage[5–7], separation[8–11], catalysis[12,13], sensing[14–16] and capture of toxic chemicals[17,18]. Among the possible gas storage applications, a potentially game-changing use that has not been explored enough is their ability to store oxygen. There is a significant need for oxygen storage in the healthcare domain, particularly in oxygen tanks for patients with respiratory disorders, as well as first aid responders. Oxygen storage is also in high demand for military and industrial applications[19,20]. Adsorption-based storage permits oxygen to be stored at much lower pressure and in much higher amounts than an empty tank, providing safer, lighter and more cost-effective alternative to high-pressure (ca. 140 bar) tanks. Other porous materials such as activated carbons and zeolites have been used for oxygen storage, but their limited tunability and relatively low capacity are major drawbacks[20–22]. However, these properties can be readily engineered in MOFs, allowing design of materials with improved performance.

A major question that often arises with the application of MOFs is how to find the top structures for a given application in the diverse pool of existing structures; indeed, in a recent collaboration with the Cambridge Crystallographic Database Centre (CCDC), we analyzed the Cambridge Structural Database—i.e., the world's repository of crystalline materials—and found that ca. 82,000 different MOFs have been published so far[1]. In efforts to address this question, a number of studies have focused on high-throughput computational screening of hypothetical MOFs for various applications[23–27]. In particular, DeCoste et al.[20] performed oxygen adsorption simulations in 10,000 hypothetical MOFs, finding NU-125[28] as the top candidate—a structure that had been synthesized before in their lab—and they confirmed the results by experimental synthesis and adsorption measurements[20]. While computational screening of hypothetical structures is clearly useful to identify top-performing candidates, in reality it is not always straightforward to synthesize a newly proposed MOF. On the other hand, screening MOFs that have already been made in the lab has the significant advantage that the synthesis method is already known, which can greatly expedite the process towards producing promising candidates[29,30].

Here, we employ high-throughput screening (HTS) techniques through GCMC simulations to explore oxygen storage in a database of 2,932 existing MOFs previously developed by Sholl, Snurr and co-workers[29,31]. In this database, the partial atomic charges of the MOFs were accurately calculated from plane-wave density functional theory (DFT) calculations[31]. We delimit the optimal structural properties to achieve optimal oxygen

deliverable capacity in MOFs at different pressures. More interestingly, while data from HTS of materials are often presented in static snapshots of complex structure–property relationships, the key advantage of our HTS approach is that we analyze the results interactively and obtain structure–property relationships through 5D visualization techniques, including animations, which provide invaluable insights to guide synthetic efforts and to reveal the physical limits of performance. Obviously, this can only be achieved through exploration of a wide range of structures containing diverse textural properties. More specifically, we demonstrate here how HTS can significantly accelerate materials discovery when combined with experimental efforts. The identification of outstanding materials from the computational screening allowed us to synthesize and test the top MOF material (UMCM-152)[32], which delivers 22.5% more oxygen than the current record holding material for oxygen storage, to the best of our knowledge[20]. The measured oxygen isotherm on UMCM-152 is in excellent agreement with our predictions, which demonstrates the accuracy of our computational screening approach.

## Results

**High-throughput computational screening of MOFs.** For oxygen tank storage, an ideal MOF structure should not only have high oxygen storage capacity, but more importantly should possess high deliverable capacity. We define the deliverable capacity here as the difference between the amount of oxygen adsorbed at the storage pressure of 140 bar and the release pressure of 5 bar. Therefore an ideal MOF material should maximize adsorption at storage pressure and minimize oxygen adsorption at low pressure. Similar to the storage of other gases such as methane[33] and hydrogen[5,34], the oxygen deliverable capacity likely depends on a number of structural properties including pore size, void space, density of the framework and the heat of adsorption. To maximize oxygen deliverable capacity, the combination of these factors must be optimized. Figure 1 shows the relationship between oxygen deliverable capacity at 298 K, oxygen heat of adsorption, and MOF void fraction. The maximum oxygen gravimetric capacity obtained is ca. 20 mol kg$^{-1}$ and is achieved for structures with heats of adsorption between 9–14 kJ mol$^{-1}$, whose void fractions are around 0.8 (i.e., corresponding to large pore volumes), as shown by the dark blue and purple data points in Fig. 1a. The mild range of heats of adsorption for oxygen is beneficial for heat management during the adsorption/desorption processes. Although stronger affinity between oxygen and MOF framework is intuitively desired, heats of adsorption higher than 16 kJ mol$^{-1}$ are only obtained for materials with void fractions

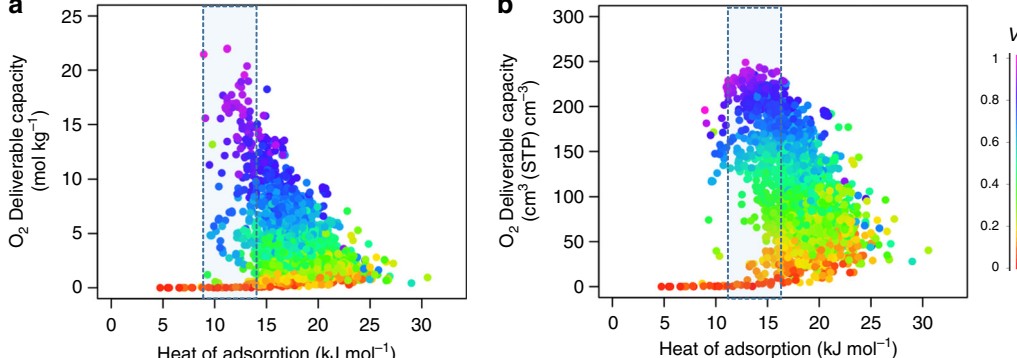

**Fig. 1** Structure–property relationships for oxygen storage in MOFs. Oxygen **a** gravimetric and **b** volumetric deliverable capacity (at 140 bar storage and 5 bar release pressures) vs. heat of adsorption for 2,932 MOFs. Each point in the graph represents a different structure. The data points are color coded by void fraction ($V_f$). See http://aam.ceb.cam.ac.uk/mof-explorer for interactive structure–property graphs

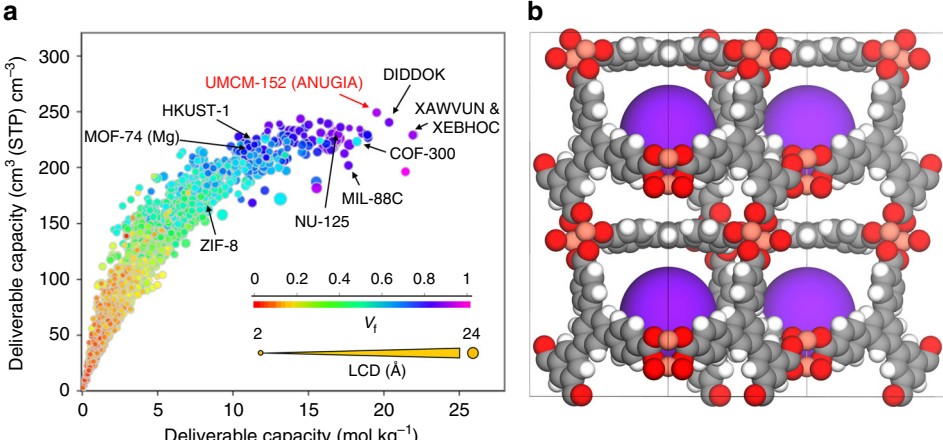

**Fig. 2** Top-performing materials for oxygen storage. **a** Oxygen volumetric and gravimetric deliverable capacities (at 140 bar storage and 5 bar release pressures) for 2932 MOF structures at 298 K. Each point in the graph represents a different structure. The data points are color coded by void fraction ($V_f$). Arrows represent common MOFs as well as promising materials identified in this work. **b** The crystal structure (super cell $2 \times 2 \times 1$) for the top material identified for volumetric oxygen storage, UMCM-152. Main cavity is represented by purple sphere

**Table 1 Top 10 structures identified for volumetric O₂ storage along with their calculated textural properties and deliverable capacities obtained at 140 bar storage and 5 bar release pressures at 298 K**

| CCDC ref code | LCD (Å) | PLD (Å) | Surface area ($m^2\,g^{-1}$) | Density (g $cm^{-3}$) | Void fraction (—) | Volumetric deliverable capacity ($cm^3$(STP) $cm^{-3}$) | Gravimetric deliverable capacity (mol $kg^{-1}$) |
|---|---|---|---|---|---|---|---|
| ANUGIA (UMCM-152) | 13.8 | 6.8 | 3760 | 0.57 | 0.86 | 249 | 19.6 |
| MOCKAR | 10.8 | 6.7 | 2948 | 0.75 | 0.83 | 243 | 14.5 |
| ANUGUM | 8.2 | 7.0 | 3444 | 0.68 | 0.81 | 241 | 15.8 |
| DIDDOK | 9.6 | 8.2 | 4639 | 0.53 | 0.83 | 240 | 20.4 |
| BICDAU | 11.5 | 6.7 | 3557 | 0.65 | 0.84 | 239 | 16.3 |
| HIHNUJ | 8.2 | 8.0 | 2772 | 0.81 | 0.83 | 239 | 13.1 |
| HIGRIA | 11.3 | 6.9 | 3477 | 0.65 | 0.84 | 239 | 16.4 |
| KEFBEE | 11.1 | 7.0 | 3087 | 0.70 | 0.82 | 238 | 15.0 |
| WEBKOF | 9.6 | 6.5 | 2592 | 0.85 | 0.80 | 237 | 12.4 |
| ICALOP | 7.5 | 6.0 | 2908 | 0.80 | 0.82 | 236 | 13.2 |

*LCD* largest cavity diameter, *PLD* pore-limiting diameter

smaller than 0.8, which corresponds to smaller pore volumes and in turn poorer deliverable capacities. Due to limited volume in storage tanks, the volumetric oxygen deliverable capacity—directly related to the size of the tank—is more crucial than gravimetric targets. The maximum volumetric oxygen capacity reaches ca. 249 $cm^3$ (STP) $cm^{-3}$ and is found for structures with adsorption heats around 11–16 kJ $mol^{-1}$ (Fig. 1b).

In order to improve the analysis of the generated large data sets, we developed an interactive 5D visualization tool for comprehensive data mining and to convey the full information obtained on O₂ storage. This is accesible at http://aam.ceb.cam.ac. uk/mof-explorer. This tool allows one to visualize O₂ gravimetric and volumetric uptakes and deliverable capacities with respect to different combinations of structural properties such as void fraction, largest cavity diameter (LCD), pore-limiting diameter (PLD), isosteric heat of adsorption, and surface area to better understand their role in O₂ adsorption performance, while allowing the user to visualize up to 5 dimensions simultaneously. Furthermore, instead of anonymous symbols, each datapoint (i.e., each MOF) can be individually identified, along with its specific properties, and tracked at different pressures. Figure 1 shows a small number of structures (CSD codes: BEKSAM, JAVTAC, GOMRAC, CUNXIS, GIHBII, GOMREG, MECLEL, QUFGIH,

LOFZUB, VAHSIH, PARMIG, MIMVEJ, and COYTOA) with relatively high void fractions and lower-than-expected deliverable capacities. All these structures show high heat of adsorption values, and when representing absolute uptake instead of deliverable capacity, they fit in the general trend. Following this idea, Supplementary Fig.1 shows the comparison of volumetric deliverable capacity versus uptake and clearly shows that these "outliers" are the MOFs that deviate more from the linearity, and that the deviation is strongly correlated with the elevated heat of adsorption values.

Figure 2a is a snapshot of the web tool capability, showing the relationship between volumetric and gravimetric oxygen deliverable capacity for all structures in the database while highlighting a number of well-known MOFs and top-performing materials; void fraction and LCD are included as a third (color) and fourth (size) dimension, respectively. One important aspect of this relationship is the emergence of a trade-off between volumetric and gravimetric uptakes similar to what has been previously observed for methane and hydrogen adsorption in MOFs[5,33,34]. The volumetric oxygen uptake reaches a maximum at ca. 250 $cm^3$ (STP) $cm^{-3}$ and then starts to plateau for those structures having large cavities (e.g., >10 Å) and void fractions (e.g., >0.8). Unlike volumetric deliverable capacity, the gravimetric deliverable

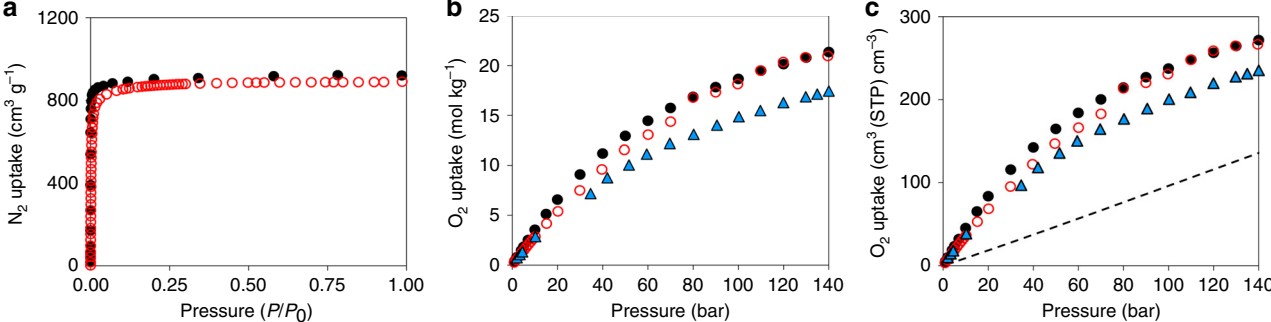

**Fig. 3** Comparison of experimental and simulated nitrogen and oxygen uptake for UMCM-152. **a** Nitrogen adsorption isotherms obtained at 77 K: black circles, simulation; red open circles, experiments. **b** gravimetric and **c** volumetric oxygen adsorption isotherms at 298 K. The experimental isotherms for NU-125 are included for comparison (blue triangles)[20]. The dashed line indicates adsorption in an empty tank at 298 K

capacity increases in structures with large cavities and void fractions (i.e., materials with low framework densities). We remark here that two considerations must be taken into account when screening the 2932 structures present in the studied database. First, this database contains ca. 75% of the MOF materials reported in the original CoRE MOF database[29]. This is due to the fact that the charge density calculations for the large structures with primitive cells of several hundred atoms are computationally intensive and challenging; therefore, some of the large-unit cell and large-pore MOFs are not present in this database. Second, DDEC and CoRE MOF databases contain MOF materials published prior to 2014. While MOFs containing large pore volumes and surface areas show higher gravimetric adsorption capacities, their typical low densities will translate into very low volumetric capacities.

Table 1 compares the top ten structures that emerged from our screening for volumetric $O_2$ deliverable capacity; Supplementary Fig. 2 shows their crystal structures. Common textural properties of these promising MOFs are LCDs of more than 7.5–8 Å, void fractions larger than 0.8 and geometric surface areas larger than 2600 $m^2 g^{-1}$. It is worth highlighting that among MOFs exhibiting high volumetric deliverable capacities, those materials with higher densities demonstrate smaller gravimetric oxygen deliverable capacities (see HIHNUJ, WEBKOF and ICALOP in Table 1). Our HTS predicts that the best material for volumetric oxygen storage is ANUGIA (UMCM-152)[32] with 249 $cm^3$(STP) $cm^{-3}$ deliverable capacity; this structure is also among the top four materials for gravimetric oxygen storage, with 19.6 mol $kg^{-1}$. Remarkably, this is a 22.5% increase over the best previously reported value for NU-125[20], and 144 and 196% improvement over gravimetric deliverable capacities measured for Norit activated carbon and zeolite NaX, respectively[20]. Here, we emphasize again on the existance of a trade-off between volumetric and gravimetric adsorption upatkes. For example while the large porosity of Al-soc-MOF-1[19], one of the top materials for oxygen storage found in the literature and not included in the DDEC database, presents a very high gravimetric deliverable capacity of 26.5 mol $kg^{-1}$ of oxygen, its volumetric deliverable capacity is ca. 202 $cm^3$(STP) $cm^{-3}$, i.e., 22.8% lower than that of UMCM-152. Figure 2b shows the UMCM-152 structure, which consists of Cu-Cu paddlewheel units connected through tetracarboxylated triphenylbenzene linkers forming two types of pores. Supplementary Fig.3 shows the simulated, geometric pore size distribution of UMCM-52 and the two main cavities with diameters of 13.8 Å and 11.4 Å, respectively.

**Synthesis of the top structure and $O_2$ adsorption measurements.** Encouraged by the HTS results, we synthesized UMCM-152 and measured experimentally its performance for $O_2$ adsorption. Given that the adsorption simulations were carried out in perfect crystals (i.e., no structural defects and no solvent molecules in the pores), experimental synthesis and activation of UMCM-152 were crucial steps towards making the optimum material. Therefore, after the MOF synthesis we first exchanged the synthesis solvent with anhydrous ethanol after washing steps, and utilized supercritical $CO_2$ activation to remove the ethanol from the pores. The MOF was handled in an argon filled glovebox thereafter. Prior to collecting $N_2$ and $O_2$ isotherms, we heated the material under vacuum to remove any possible residual moisture. Comparison of measured and simulated $N_2$ adsorption isotherms at 77 K demonstrated excellent agreement at saturation loading, indicating that the synthesized sample was highly crystalline and successfully activated (Fig. 3a and Supplementary Fig. 4). The guest free UMCM-152 samples were then subjected to high-pressure oxygen adsorption to examine their storage capacity at room temperature. Figure 3b-c shows the comparison of measured and simulated oxygen adsorption isotherms at 298 K. The simulated gravimetric and volumetric $O_2$ isotherms match the experimental isotherms well, confirming the validity of our HTS approach; the difference between predicted and experimental $O_2$ deliverable capacity is <0.2 % between 140 bar and 5 bar. As predicted, gravimetric and volumetric deliverable capacities exceed those of the best material known to date, NU-125 (blue triangles), by 22.5 and 15%, respectively.

The comparison between the volumetric adsorption in UMCM-152 and an empty tank in Fig. 3c demonstrates how this material can tremendously boost the storage capacity of oxygen, with 96% improvement in the amount adsorbed in UMCM-152 (266 $cm^3$ (STP) $cm^{-3}$) compared to conventional compressed oxygen storage at 140 bar (136 $cm^3$ (STP) $cm^{-3}$). It is worth mentioning that one would have to compress oxygen up to 300 bar in an empty tank to reach oxygen densities offered by a UMCM-152-packed tank at only 140 bar. This is particularly attractive for making storage systems lighter, smaller and safer. Superior volumetric and gravimetric oxygen deliverable capacities are not the only factors that need to be taken into account for practical materials discovery. To ultimately identify optimal adsorbents, heat management due to adsorption and desorption of gases, efficient packing of the adsorbent into a tank, stability toward impurities such as water, recyclability, as well as adsorbent cost need to be included in any gas storage application. In this context, powerful dynamic visualization tools play a crucial role in efficiently exploring the different possibilities in the structure–property landscape and in accurately pinpointing MOFs with specific desired properties. This could include, for example, those synthesized from commercially available linkers such as benzene 1,4-dicarboxylic acid or benzene-1,3,5-tricarboxylic acid.

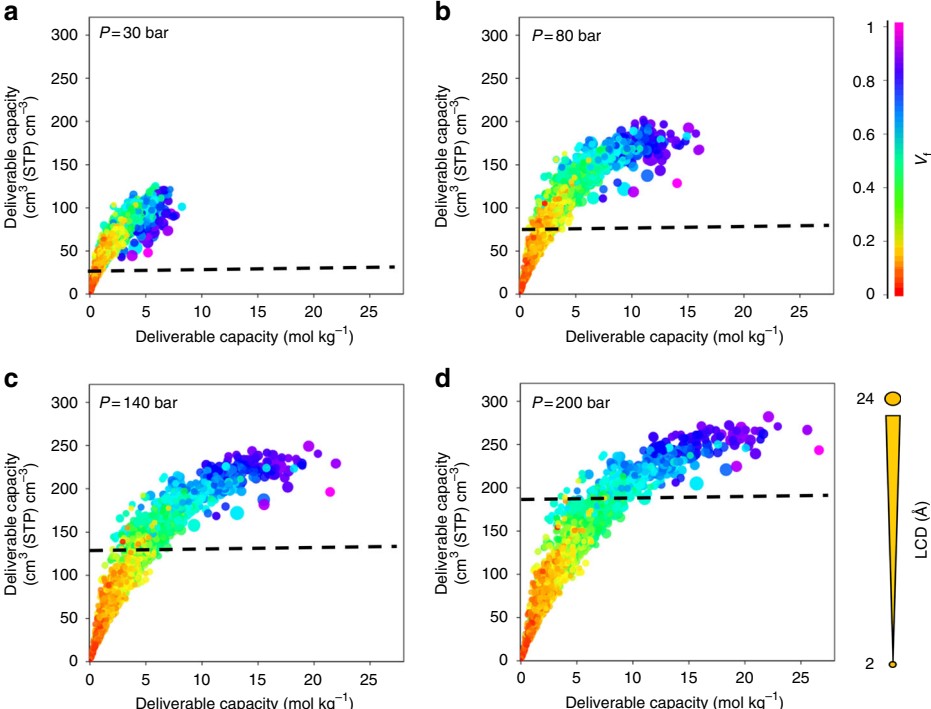

**Fig. 4** Structure–property relationships for oxygen storage in MOFs. Oxygen volumetric and gravimetric deliverable capacity is plotted vs. the largest cavity diameter (LCD) and void fraction ($V_f$) for 2,932 MOF structures at **a** 30 bar, **b** 80 bar, **c** 140 bar and **d** 200 bar storage pressures and 298 K. The release pressure is kept fixed at 5 bar for all storage pressures. The dashed lines mark the amount of oxygen adsobed in an empty tank. Each point in the graph represents a different structure. The data points are color coded and sized according to $V_f$ and LCD, respectively

**5D interactive visualization and data mining**. In order to explore the limits of MOFs for storing oxygen in a wider range of pressures, we also analyzed the oxygen adsorption at storage pressures of 1, 5, 10, 20, 30, 50, 80, 100, 140 and 200 bar for all the MOFs in the database. Our MOF explorer tool (http://aam.ceb.cam.ac.uk/mof-explorer) gives access to the interactive interface (see Supplementary Movie 1). Note that for all the generated data sets, users can now not only visualize different combinations of structure–property relationships in 5D but also pinpoint and track any material's performance interactively as oxygen storage pressure increases. Within the MOF explorer, it is also possible to narrow down and filter the MOF database looking for a specific range of structures and properties including "outliers". For all pressure points combined, over 1000 unique structure–property plots can be generated according to the user's interest. Figure 4 compares snapshots of oxygen deliverable capacities obtained for selected storage pressures of 30, 80, 140 and 200 bar, keeping the release pressure fixed at 5 bar; Supplementary Fig.5–7 show different snapshots for the representation of the evolution of performance in the adsorption process at different pressures. The deliverable capacity in MOFs with pore sizes less than 5 Å and void fractions less than ca. 0.3 (see yellow and red data points) does not change much from 30 to 200 bar as their pores become saturated with oxygen at low pressures. For all storage pressures, the maximum volumetric and gravimetric oxygen deliverable capacities are obtained for structures with LCD values larger than ca. 7 Å and void fractions larger than ca. 0.6 (see also Supplementary Fig. 7). These optimum peaks shift slightly towards larger pores as the storage pressure increases. For MOFs with void fractions larger than 0.8, the amount of oxygen adsorbed at 5 bar is very low and the amount adsorbed at high pressures (e.g., 140 or 200 bar) is near saturation. Therefore, at these elevated pressures, the deliverable capacity is mainly determined by the storage capacity (Supplementary Fig. 8). The highest volumetric

deliverable capacities achieved for 30 and 80 bar are ca. 125 and 200 cm³ (STP) cm⁻³, respectively (empty tank: 23 and 72 cm³ (STP) cm⁻³, respectively), compared to ca. 250 cm³ (STP) cm⁻³ obtained at 140 bar (empty tank: 131 cm³ (STP) cm⁻³). At 200 bar, the maximum volumetric deliverable capacity is ca. 282 cm³ (STP) cm⁻³, compared with an empty tank deliverable capacity of 189 cm³ (STP) cm⁻³. Using the online MOF explorer tool, one can also probe the effects of oxygen desorption pressure and monitor how the deliverable capacity of individual MOFs changes with respect to release pressure; for example, the deliverable capacity for UMCM-152 is increased from 250 cm³ (STP) cm⁻³ to 270 cm³ (STP) cm⁻³ when the release pressure is decreased from 5 bar to 1 bar and the storage pressure is kept fixed at 140 bar.

The "sweet spot" combining different geometrical properties with optimum uptake or deliverable capacity changes as the storage pressure increases from 30 to 200 bar. For example, top volumetric uptake at 140 bar (i.e., higher than 250 cm³ (STP) cm⁻³) is obtained for MOFs with void fractions between 0.45 and 0.82, but in this range of void fractions, the volumetric uptake can be as low as 125 cm³ (STP) cm⁻³ (Supplementary Fig. 9). To find what descriptors are more important, Fig. 5 compares the role of different MOF properties for the top 1% of structures – in terms of volumetric deliverable capacity – at 30, 80, 140 and 200 bar storage pressure. Supplementary Table 2 also highlights the first quartile, third quartile, and interquartile range (IQR) values for the top materials geometric properties. The general trend shows that the values of optimal LCDs, void fractions and surface areas increase, while density decreases, with the storage pressure; in general, larger pore sizes translate into higher void fractions and surface areas, as well as lower densities. The variation of individual geometric properties spread in the IQR (i.e., the box height) and how the data is skewed is particularly interesting. Remarkably, the interquartile height of LCD increases with the storage pressure, whereas those for void fraction and surface area

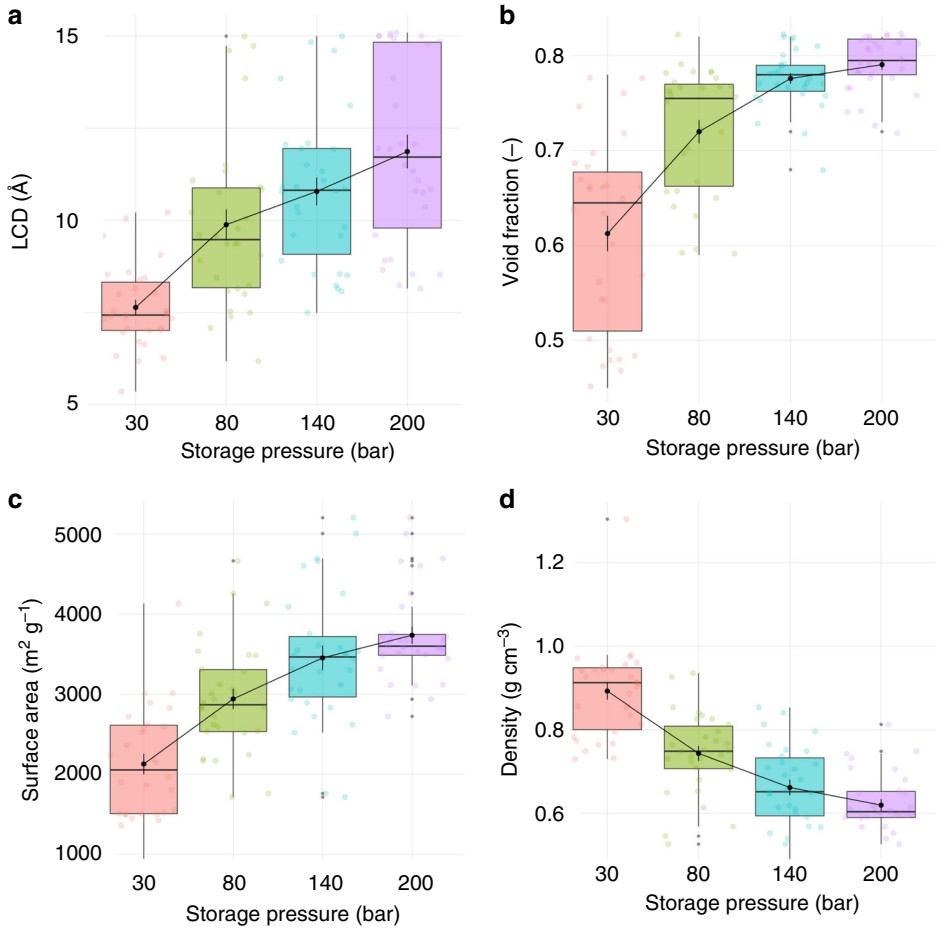

**Fig. 5** Optimal geometric properties for the top 1% of MOF structures. Box and whisker plots representing **a** largest cavity diameter (LCD), **b** void fraction, **c** surface area and **d** density, for storage pressures of 30, 80, 140 and 200 bar. The markers represent the minimum, first quartile, median, third quartile, and maximum values, respectively. Outliers, identified as 1.5 × the minimum or maximum values, are represented by gray data points. Mean values for different pressures are connected via lines with bars representing standard deviation. Data points are offset laterally for better visualization

decrease, with not so significant variations for density. This indicates the importance of optimal values of LCD at low storage pressure, and void fraction and surface area at high storage pressure. Overall, at 30 bar, optimal MOFs feature a narrow range for optimal pore size (i.e., LCD: 7.0–8.3 Å), but broader for void fraction (0.5–0.7) and surface area (1500–2600 $m^2 g^{-1}$); see Supplementary Table 2. As the storage pressure increases to 200 bar, these sweet spots change towards larger values, while the range of optimal values for LCD is now broader (LCD: 9.8–14.8 Å), but much narrower for void fractions (0.78–0.82) and surface area (3500–3700 $m^2 g^{-1}$). This clearly indicates that top materials at a certain storage pressure do not necessarily keep the same ranking as the storage pressure changes. For example, the top structures at 30 bar, HAFQOW and EHALOP, are ranked 280[th] and 278[th], respectively, at 140 bar.

In conclusion, with advances in synthetic methods, thousands of functional materials including MOFs are made every year, bringing a new challenge to find which materials are most suitable for a given application. In this study, we have shown how a systematic large-scale computational screening of a database of known MOFs can guide synthetic efforts towards promising materials and dramatically accelerate materials discovery. We developed a visualization interface that can support analysis of large HTS data and dynamically delimit 5D structure–property relations. We have made this interface and all visualization features publically available. Through molecular-level simulations

of 2932 structures, we showed $O_2$ adsorption limits in MOFs, delineated how structural properties affect $O_2$ deliverable capacity and—in a rare computer-aided case study—discovered a MOF material (UMCM-152) that is able to deliver 22.5% more oxygen than the best material known to date and to improve oxygen deliverable capacity by 90% over storage in an empty tank.

## Methods

**Computational methods**. The grand canonical Monte Carlo (GCMC) simulations of oxygen adsorption were carried out at 298 K and at pressures of 1, 5, 10, 20, 30, 50, 80, 100, 140 and 200 bar. The oxygen–MOF and oxygen–oxygen interactions were modeled using a Lennard–Jones (LJ) plus Coulomb potential. The LJ potential was cut-off at 12.8 Å. All electrostatic interactions were calculated using the Ewald summation method. The force field parameters for oxygen and nitrogen were taken from the TraPPE force field (Supplementary Table 1). The LJ parameters for the framework atoms were taken from the Universal Force Field. To ensure accuracy in our calculations, we performed our simulations on a subset of 2,932 MOF structures with high-quality atomic point charges for the structures developed by Nazarian et al. using periodic DFT calculations[31]. The DDEC partial charges properly reproduce the electrostatic potential in the MOF pores and hence provide an accurate representation of electrostatic interactions between the MOF and the adsorbates with polar and quadrupolar interactions. All frameworks were considered as rigid during the simulations. Insertions, deletions, translations and rotation Monte Carlo moves were attempted in the simulation cell. For all pressure points, we used 5000 cycles for equilibration and 5000 cycles to average properties. A cycle is defined as the maximum of 20 or the number of molecules in the system. All calculations were carried out in the RASPA molecular simulation software[35].

**5D visualization platform**. All of the graphs presented in this paper can be reproduced online at http://aam.ceb.cam.ac.uk/mof-explorer. Furthermore, visitors to the site can explore the entire data set interactively, with any one of 11 variables plotted on each of the five axes. Since data has been gathered at multiple pressure points, this leads to over 1000 unique plots which can be generated according to the user's interest. MOFs can be searched for and filtered either by name or by property, or by selecting them from the graph, allowing the user to track a particular MOF's characteristics through changing pressure (see Supplementary Movie 1).

**UMCM-152 synthesis**. UMCM-152 was synthesized according to literature[32] with some modifications. In a 8 dram vial, 50 mg of linker **4** (0.10 mmol) (Supplementary Fig. 12) was dissolved in a Dimethylformamide (DMF)/dioxane/$H_2O$ (4:1:1, 10 mL) mixture; 50 μL of 1 M aq. HCl solution was added. To this mixture, 96 mg of $Cu(NO_3)_2 \cdot 2.5H_2O$ (0.41 mmol) was added and the contents were sonicated until dissolved. The resulting solution was divided into 10 of 1.5 dram vials equally and heated to 85 °C. (Allowing the UMCM-152 crystals to grow more than 2 h resulted in the formation of significant amounts of light blue impurities.) After 2 h the mother solution was pipetted out, and the dark blue crystals were washed with fresh DMF three times. In some vials minimal amount of fluffy light blue impurities were observed, which can easily be removed by a glass pipet after adding fresh DMF. After washing with DMF, the crystals were washed with absolute ethanol three times and soaked in absolute ethanol for 18 h. Ethanol was removed by supercritical $CO_2$ activation and then the sample was transferred into the surface area analyzer tube using a glove box filled with argon.

**Oxygen adsorption measurements**. UMCM-152 was analyzed using a HPVA-II 200 from Micromeritics/Particulate Systems to determine the adsorption capacity of $O_2$ up to 140 bar at 25 °C. Approximately 100 mg of UMCM-152 was loaded into the sample chamber in an Ar backfilled glove box. Prior to analysis vacuum was applied to the sample as it was heated to 60 °C for two hours and then ramped to 100 °C for an additional two hours.

**Data availability**. Details of the computational and experimental methods are outlined in Supplementary Method 1 & 2. Structure–property graphs can be viewed online at http://aam.ceb.cam.ac.uk/mof-explorer. All structures in the website are linked to the Cambridge Structural Database (CSD) for easier access and analysis. Any additional experimental/simulation data set generated and/or analyzed during the current study are available from the corresponding authors on reasonable request.

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

## Acknowledgements

D.F.-J. thanks the Royal Society for funding through a University Research Fellowship and the European Research Council (ERC) under the European Union's Horizon 2020 research and innovation programme (NanoMOFdeli), ERC-2016-COG 726380. R. Q. S. and O. K. F. acknowledge financial support from the U.S. Department of Energy (grant number DE-FG02–08ER15967). C. F. K. acknowledges funding from the UK Engineering and Physical Sciences Research Council, EPSRC (grants EP/L015889/1 and EP/H018301/1), the Wellcome Trust (grants 3–3249/Z/16/Z and 089703/Z/09/Z), the UK Medical Research Council, MRC (grants MR/K015850/1 and MR/K02292X/1), MedImmune, and Infinitus (China) Ltd. Computational work was supported by the Cambridge High Performance Computing Cluster, Darwin. We also thank Aurelia Li for useful discussions.

## Author contributions

P. Z. M. and D. F.-J. designed the research. P. Z. M. drafted the manuscript, performed geometric characterization of structures, carried out GCMC simulations and analyzed the isotherms; T. I. carried out the synthesis, activation and handling of UMCM-152 under the supervision of O. K. F.; S. G. carried out the synthesis of the UMCM-152 linker under the supervision of O. K. F.; J. E. performed $O_2$ adsorption measurements; M.F. developed

MOF explorer under supervision of C.F.K.; P. Z. M.; R. Q. S.; O. K. F.; and D. F-J. analyzed the data and all authors contributed to the editing of the manuscript.

## Additional information

**Competing interests:** D.F.-J. has financial interest in the start-up company Immaterial Labs, which is seeking to commercialize metal–organic frameworks. The remaining authors declare no competing interests.

