## [Peer Review File · Nature Communications]

Reviewer #1 (Remarks to the Author):

Computer-Aided Discovery of a Metal-Organic Framework with Superior Oxygen uptake

Recommendation. Herein, P. Z. Moghadam et al. discover a new optimal material for oxygen storage by maximizing known descriptors in the previously unexplored family of metal organic frameworks. Additionally, a tool is created and made publically available for exploring trends among these materials in the context of up to five variables simultaneously. Though the merits of such deliverables are substantial, additional discussion/analysis is requested that is standard to such investigations. With these points addressed, I recommend the article for publication in Nature Communications.

Article Outline. 10,000 metal organic frameworks (MOF) from the Cambridge Structural Database are explored for oxygen adsorption. In particular, MOFs have not been previously considered for this application despite being easily engineered for specific applications and, thus, potentially more attractive than industry standards. Grand canonical Monte Carlo simulations are employed to explore oxygen storage among these materials, and plane-wave density functional theory calculations for the calculation of partial atomic charges. Several structure-property relationships are highlighted in the paper – in particular, oxygen gravimetric and volumetric deliverable capacity vs. heat of adsorption in the context of void fraction. Changes of performance with pressure are also considered. Top performers occupy a distinct cross section of these features, which offers an excellent screening opportunity. UMCM-152 appears at the top of structures identified for volumetric oxygen storage, which is directly related to storage tank size. The material is synthesized and tested, exhibiting a performance that matches well with the aforementioned calculations and delivering 22.5% more oxygen than the best known material to date. Finally, a public domain web application has been created for plotting these structure-property relationships – offering opportunities for future investigations to leverage these findings.

Revisions. Please address the following:

What is the practical impact of such a discovery? Optimizing the desired property is not the only consideration for utility. Synthesis is clearly possible, but is it feasible/scalable? Speak specifically to UMCM-152.

Descriptors discussed herein are known to impact adsorption of oxygen. a) Which are most important, b) are these limited to this domain (MOF), and c) could there be others? In general, these are addressed through a quantitative/statistical analysis of the importances of these descriptors. This goes toward the computer-aided component of this investigation.

The trends explored herein are qualitatively strong (e.g., Figure 1), but there are clear outliers (see bottom right corners of plots). The strength of these trends should be quantified (Point 2), and some discussion of outliers should be added. Are these outliers not important at all?

Minor Points.

The paper notes that structure-property plots show up to four dimensions. I believe there are really five (three spatial, color, and point size) unless I understood incorrectly.

It would be helpful to list the meaning of each abbreviation within the online tool (LCD, PLD, etc.). The full name comes up on the graph, but only after making the selection.

Including meta-data of selected compounds would also be helpful (perhaps in a separate table upon clicking).

Reviewer #2 (Remarks to the Author):

The paper reports a Computer-aided strategy to find a Metal-Organic Framework adsorbents with high O₂ uptake.

One of the predicted structures was isolated experimentally with big discrepancy in O₂ uptake between the simulated and the experimentally made structure. The reason of this discrepancy is

not discussed rationally.

The authors claim that they explored theoretically diverse structural properties but pore size is not the only one parameter affecting the Storage of gases. If that is the case then the same conclusion for O₂ should be applied for H₂, CH₄ and CO₂. Not only the pore size is important but also the charge density (functionality), the Pore size distribution, the presence of clusters or chains, topological features which make this task difficult. Some parameters should be fixed of course to make a systematic study, but in this work the study is so simplistic and can be perceived not strong.

Accordingly, I do not see really a hypothetical study about structural properties relationships that deserve a publication in Nature communication. Numerous statements in this paper about the trade of between gravimetric and volumetric uptakes, an already established finding and not a discovery in this work.

The authors defined a lower limit for oxygen at 5 bar to determine the deliverable uptake which is clearly not understood. What is the constraint or the rational in doing so?

Up to page 9 , there is a lot of claims about the "highest ", the "lowest" but there is no constructive and systematic scientific discussion about what is needed to make a better storage media for O₂.

Finally, I do not think that "4D Interactive visualization of data" adds any valuable scientific element that should be published in Nature communication Journal.

In light of the above cited elements, I cannot be positive in supporting this paper for acceptance in Nature communication.

Re: Decision on ID NCOMMS-17-21131-T

Title of the paper: Computer-Aided Discovery of a Metal-Organic Framework with Superior Oxygen Uptake

Reviewer #1

Recommendation. *Herein, P. Z. Moghadam et al. discover a new optimal material for oxygen storage by maximizing known descriptors in the previously unexplored family of metal organic frameworks. Additionally, a tool is created and made publically available for exploring trends among these materials in the context of up to five variables simultaneously. Though the merits of such deliverables are substantial, additional discussion/analysis is requested that is standard to such investigations. With these points addressed, I recommend the article for publication in Nature Communications.*

Article Outline. *10,000 metal organic frameworks (MOF) from the Cambridge Structural Database are explored for oxygen adsorption. In particular, MOFs have not been previously considered for this application despite being easily engineered for specific applications and, thus, potentially more attractive than industry standards. Grand canonical Monte Carlo simulations are employed to explore oxygen storage among these materials, and plane-wave density functional theory calculations for the calculation of partial atomic charges. Several structure-property relationships are highlighted in the paper – in particular, oxygen gravimetric and volumetric deliverable capacity vs. heat of adsorption in the context of void fraction. Changes of performance with pressure are also considered. Top performers occupy a distinct cross section of these features, which offers an excellent screening opportunity. UMCM-152 appears at the top of structures identified for volumetric oxygen storage, which is directly related to storage tank size. The material is synthesized and tested, exhibiting a performance that matches well with the aforementioned calculations and delivering 22.5% more oxygen than the best known material to date. Finally, a public domain web application has been created for plotting these structure-property relationships – offering opportunities for future investigations to leverage these findings.*

Revisions. *Please address the following:*

1. What is the practical impact of such a discovery? Optimizing the desired property is not the only consideration for utility. Synthesis is clearly possible, but is it feasible/scalable? Speak specifically to UMCM-152.

We thank the reviewer for the positive feedback and the excellent suggestions. Feasibility studies for practical gas storage settings are certainly an interesting point. As noted by the reviewer, superior volumetric and gravimetric oxygen deliverable capacities are not the only factors that need to be taken into account for materials discovery. To ultimately identify optimal adsorbents, heat management due to adsorption and desorption of gases, efficient packing of the adsorbent into a tank, stability toward impurities such as water, recyclability, as well as adsorbent cost need to be included in any gas storage application. In this sense, our simulations are important mainly as a screening process to identify promising MOFs, such as UMCM-152. In addition, following this screening, we successfully synthesised and tested UMCM-152; scalability studies as well as the other factors mentioned above are beyond the scope of this study, and indeed they constitute the basis of an entire separate study/publication about the refinement of the synthesis process. Having said that, one of the benefits of

our dynamic visualization tools is that it will allow the reader not only to focus on the MOF we identified (UMCM-152) – as it generally occurs with static figures/representations benchmarking materials – but to study any other MOF with the desired properties. This could include for example those synthesized from commercially available linkers such as benzene 1,4-dicarboxylic acid, or benzene-1,3,5-tricarboxylic acid. To emphasise on the reviewer’s suggestion, we have added the discussion mentioned above on different factors influencing oxygen storage in the manuscript:

Page 9: “However, superior volumetric and gravimetric oxygen deliverable capacities are not the only factors that need to be taken into account for practical materials discovery. To ultimately identify optimal adsorbents, heat management due to adsorption and desorption of gases, efficient packing of the adsorbent into a tank, stability toward impurities such as water, recyclability, as well as adsorbent cost need to be included in any gas storage application. In this context, powerful dynamic visualization tools play a crucial role in efficiently exploring the different possibilities in the structure-property landscape, and in accurately pinpointing MOFs with specific desired properties. This could include, for example, those synthesized from commercially available linkers such as benzene 1,4-dicarboxylic acid or benzene-1,3,5-tricarboxylic acid”.

2. Descriptors discussed herein are known to impact adsorption of oxygen. a) Which are most important, b) are these limited to this domain (MOF), and c) could there be others? In general, these are addressed through a quantitative/statistical analysis of the importances of these descriptors. This goes toward the computer-aided component of this investigation.

Oxygen deliverable capacity depends on multiple structural properties including cavity/window size, void space, surface area, framework density as well as the heat of adsorption – and to maximize the deliverable capacity, the combination of these factors must be optimized. As can be expected, the question of which descriptor is the most important is not straightforward. MOFs sharing one identical property may have different ranges of other properties, which explains the wide range of oxygen deliverable capacities among them. Addressing this exact question is the main objective of the web-based visualization platform we developed in this work for our high-throughput screening (HTS) approach; the key advantage is that it allows analysing the results interactively through over 1000 structure-property relationships 5D plots. This capability is key to provide the “combination of descriptors” that affect oxygen storage at different storage pressures. For example, we observe that the maximum oxygen gravimetric capacity at 140 bar is obtained for structures with heats of adsorption between 9-14 kJ/mol, whose void fractions are larger than 0.8, with pore cavities of more than 8 Å, and densities smaller than 1.5 g/cm³. These “sweet spots” with optimum deliverable capacity change as the storage pressure varies from 30 to 200 bar.

To fully address the reviewer’s comment, we first focus here on the dynamic representation: plots of volumetric uptake as a function of largest cavity diameter, pore limiting diameter, void fraction, density, or heat of adsorption show great dispersion. For example, top volumetric uptake at 140 bar (i.e. those MOFs showing volumetric uptake higher than 250 cm³/cm³) is obtained for MOFs with void fractions between 0.45 and 0.82 (see Supplementary Figure 9; now added to the paper). However, in this range of void fractions, the volumetric uptake can be as low as 125 cm³/cm³. In a second step, to find the most important descriptors in top materials, we have investigated the effects of the textural properties on deliverable capacity for the top 1% of structures with superior volumetric deliverable

capacities at 30, 80, 140 and 200 bar storage pressures, keeping the release pressure fixed at 5 bar – please see Figures 5 and Supplementary Table 2; both now added to the revised version. The geometrical properties of optimum materials change according to the storage pressure. Structures with superior performance at a certain storage pressure do not necessarily outperform other materials as the storage pressure changes. Although this is expected, our dynamic representation allows studying it in detail and with considerable ease. For example, at 30 bar structures HAFQOW and EHALOP are the top two materials for volumetric deliverable capacity, but they are ranked 280 and 278, respectively, at 140 bar storage pressure because of their medium sized pores of ca. 7.5 Å.

Supplementary Figure 9. Relationship between the oxygen uptake at 140 bar and void fraction for 2,932 MOF structures at 298 K. The data points are color coded and sized by heat of adsorption and largest cavity diameter, respectively. Each point in the graph represents a different structure.

We have added a discussion to the manuscript to thoroughly explain these points, Page 12: “The “sweet spot” combining different geometrical properties with optimum uptake or deliverable capacity changes as the storage pressure increases from 30 to 200 bar. For example, top volumetric uptake at 140 bar (i.e. higher than 250 cm³/cm³) is obtained for MOFs with void fractions between 0.45 and 0.82, but in this range of void fractions, the volumetric uptake can be as low as 125 cm³/cm³ (Supplementary Figure 9). To find what descriptors are more important, Figure 5 compares the role of different MOF properties for the top 1% of structures – in terms of volumetric deliverable capacity – at 30, 80, 140 and 200 bar storage pressure. Supplementary Table 2 also highlights the 1st quartile, 3rd quartile, and interquartile range (IQR) values for the top materials geometric properties. The general trend shows that the values of optimal LCDs, void fractions and surface areas increase, while density decreases, with the storage pressure; in general, larger pore sizes translate into higher void fractions and surface areas, as well as lower densities. The variation of individual geometric properties spread in the interquartile range (i.e. the box height) and how the data is skewed is particularly interesting.

Remarkably, the interquartile height of LCD increases with the storage pressure, whereas those for void fraction and surface area decrease, with not so significant variations for density. This indicates the importance of optimal values of LCD at low storage pressure, and void fraction and surface area at high storage pressure. Overall, at 30 bar, optimal MOFs feature a narrow interquartile range for optimal pore size (i.e. LCD IQR: 7.0-8.3 Å), but broader for void fraction (0.5-0.7) and surface area (1500-2600 m²/g) – see Supplementary Table 2. As the storage pressure increases to 200 bar, these sweet spots change towards larger values. While the interquartile range of optimal values for LCD is now broader (LCD IQR: 9.8-14.8 Å), it is much narrower for void fraction (0.78-0.82) and surface area (3500-3700 m²/g). This clearly indicates that top materials at a certain storage pressure do not necessarily keep the same ranking as the storage pressure changes. For example, the top structures at 30 bar, HAFQOW and EHALOP, are ranked 280th and 278th, respectively, at 140 bar.”

Figure 5. Box and whisker plots representing optimal geometric properties for the top 1% of structures as a function of storage pressure (30-200 bar): **a.** largest cavity diameter (LCD), **b.** void fraction, **c.** surface area and **d.** density. The markers represent the minimum, 1st quartile, median, 3rd quartile, and maximum values, respectively. Outliers, identified as 1.5 × the minimum or maximum values, are represented by grey data points. Mean values for different pressures are connected via lines with bars representing standard deviation. Data points are offset laterally for better visualization.

Supplementary Table 2. Geometrical property comparison between 1st quartile (Q1), 3rd quartile (Q3), and interquartile range (IQR) for the top 1% of structures as a function of storage pressure (30-200 bar).

Pressure (bar)	LCD (Å)				Surface area (m ² /g)				Void fraction (-)				Density (g/cm ³)			
	30	80	140	200	30	80	140	200	30	80	140	200	30	80	140	200
Q1	7.01	8.17	9.08	9.79	1508	2534	2965	3486	0.51	0.66	0.76	0.78	0.8	0.71	0.59	0.59
Q3	8.32	10.88	11.95	14.83	2613	3306	3720	3747	0.68	0.77	0.79	0.82	0.95	0.81	0.73	0.65
IQR	1.31	2.71	2.87	5.04	1105	772	755	261	0.17	0.11	0.03	0.04	0.15	0.1	0.14	0.06

In addition, we have included a discussion about the combinatorial effects of geometrical properties for Figure 1, Page 4: “To maximize oxygen deliverable capacity, the combination of these factors must be optimized. Figure 1 shows the relationship between oxygen deliverable capacity, oxygen heat of adsorption, and MOF void fraction. The maximum oxygen gravimetric capacity at 140 bar (ca. 20 mol/kg) is achieved for structures with heats of adsorption between 9-14 kJ/mol, whose void fractions are around 0.8 (i.e. corresponding to large pore volumes), as shown by the dark blue and purple data points in Fig. 1a. The mild heats of adsorption are particularly beneficial for heat management during the ad-/desorption processes. Interestingly, although stronger affinity between oxygen and the MOF is intuitively desired, heats of adsorption higher than 16 kJ/mol are only obtained for materials with void fractions smaller than 0.8, which corresponds to smaller pore volumes and in turn poorer deliverable capacities.”

Regarding questions 2.b and 2.c from the reviewer (*are these [descriptors] limited to this domain (MOF), and could there be others?*), given the wealth of structures and diverse properties, we believe that the principles of oxygen adsorption limits and the structure-property relationships delimited in this work are not limited to the MOF domain and can be applied to other porous materials. Regarding the existence of other descriptors, we want to highlight the fact that we used a particular database of MOFs whose partial atomic charges are calculated from accurate ab initio derived electron density distributions (i.e. Density Derived Electrostatic and Chemical (DDEC) method). Since some of these MOFs will present functional groups and therefore different surface chemistries, we are confident that the sheer number of structures probed in this report covers a large diversity, and that their effect is taken into account. Indeed, the combination of surface chemistry and textural properties will affect oxygen uptake and the heat of adsorption. Framework topology is another factor that could affect oxygen storage, but characterising the topology for ca. 3000 structures is not a trivial task and could constitute the basis for another paper.

3. The trends explored herein are qualitatively strong (e.g., Figure 1), but there are clear outliers (see bottom right corners of plots). The strength of these trends should be quantified (Point 2), and some discussion of outliers should be added. Are these outliers not important at all?

Figure 1 shows a small number of structures with relatively high void fractions and lower-than-expected deliverable capacities. Using the website, we can use the option “filter graph” to easily select these outlier structures in order to focus on them, CSD codes: BEKSAM, JAVTAC, GOMRAC, CUNXIS, GIHBII, GOMREG, MECLLEL, QUFGIH, LOFZUB, VAHSIH, PARMIG, MIMVEJ, and COYTOA. All these structures show high heat of adsorption values, and when representing absolute

uptake instead of working capacity, they fit in the general trend. Following this idea, Supplementary Figure 1 (added to the revised version) shows the comparison of volumetric deliverable capacity versus uptake. It is clear that these “outliers” are the MOFs that deviate more from the linearity, and that the deviation is strongly correlated with the heat of adsorption. We would like to further emphasize the strength of our interactive web-domain visualization tool where users can interactively identify materials – including outliers – from any region of the datasets to probe structural information.

We think this is an interesting point and we added a short discussion on the presence of outliers in the Figure 1 discussion. Page 5 first paragraph: “Figure 1 shows a small number of structures (CSD codes: BEKSAM, JAVTAC, GOMRAC, CUNXIS, GIHBII, GOMREG, MECLEL, QUFGIH, LOFZUB, VAHSIH, PARMIG, MIMVEJ, and COYTOA) with relatively high void fractions and lower-than-expected deliverable capacities. All these structures show high heat of adsorption values, and when representing absolute uptake instead of working capacity, they fit in the general trend. Following this idea, Supplementary Figure 1 shows the comparison of volumetric deliverable capacity versus uptake and clearly shows that these “outliers” are the MOFs that deviate more from the linearity, and that the deviation is strongly correlated with the elevated heat of adsorption values.”

Supplementary Figure 1. Relationship between the oxygen deliverable capacity and oxygen uptake at 140 bar for 2,932 MOF structures at 298 K. The data points are color coded and sized by heat of adsorption and void fraction, respectively. Each point in the graph represents a different structure.

We also emphasised this point on Page 10: “Within the MOF explorer, it is also possible to narrow down and filter the MOF database looking for a specific range of structures and properties including potential outliers.”

Minor Points.

The paper notes that structure-property plots show up to four dimensions. I believe there are really five (three spatial, color, and point size) unless I understood incorrectly.

It would be helpful to list the meaning of each abbreviation within the online tool (LCD, PLD, etc.).

The full name comes up on the graph, but only after making the selection.

Including meta-data of selected compounds would also be helpful (perhaps in a separate table upon clicking).

We thank the reviewer for pointing this out and the overall constructive comments. We completely agree with these minor points and we have now implemented these improvements to our MOF-explorer 5D visualization tool and throughout the manuscript.

Reviewer #2:

1. The paper reports a Computer-aided strategy to find a Metal-Organic Framework adsorbents with high O₂ uptake.

One of the predicted structures was isolated experimentally with big discrepancy in O₂ uptake between the simulated and the experimentally made structure. The reason of this discrepancy is not discussed rationally.

We do not understand the reviewer's comment about a "big discrepancy in O₂ uptake between the simulated and the experimentally made structure." The simulated and experimental O₂ isotherms in Figure 3b or 3c (black and red circles, respectively) are in excellent agreement (probably within the experimental error). Perhaps the reviewer mistakenly compared with the blue triangles, which are for a different MOF. We also would like to emphasize the fact that we choose to synthesize and test UMCM-152 solely because our HTS predicted this MOF to have the highest volumetric deliverable capacity among all ca. 3000 materials studied. This highlights the predictive capabilities of molecular simulation towards discovery of outstanding materials for applications such as oxygen gas storage. We have added more information on the comparison between experimental and theoretical isotherms on Page 8 to clarify this point and avoid any possible confusion.

Page 8: "Figure 3 b-c shows the comparison of measured and simulated oxygen adsorption isotherms at 298 K. The simulated gravimetric and volumetric O₂ isotherms match the experimental isotherms well, confirming the validity of our HTS approach to identify top-performing materials; the difference between simulated and experimental O₂ deliverable capacity is < 0.2 % between 140 bar and 5 bar. As predicted, gravimetric and volumetric deliverable capacities exceed those of the best material known to date, NU-125 (blue triangles), by 22.5% and 15%, respectively."

2. The authors claim that they explored theoretically diverse structural properties but pore size is not the only one parameter affecting the Storage of gases. If that is the case then the same conclusion for O₂ should be applied for H₂, CH₄ and CO₂. Not only the pore size is important but also the charge density (functionality), the Pore size distribution, the presence of clusters or chains, topological features which make this task difficult. Some parameters should be fixed of course to make a systematic study, but in this work the study is so simplistic and can be perceived not strong.

Accordingly, I do not see really a hypothetical study about structural properties relationships that deserve a publication in Nature communication. Numerous statements in this paper about the trade of between gravimetric and volumetric uptakes, an already established finding and not a discovery in this work.

Finally, I do not think that “4D Interactive visualization of data” adds any valuable scientific element that should be published in Nature communication Journal.

In light of the above cited elements, I cannot be positive in supporting this paper for acceptance in Nature communication.

We respectfully disagree with the overall response from the reviewer. First of all, we never claimed pore size to be “the only one parameter affecting the storage of gases”. As described in the manuscript and detailed in this response, a MOF’s performance depends simultaneously on different textural properties such as cavity and window diameter, void fraction, surface area, density as well as heat of adsorption. In addition, materials sharing only one specific property may exhibit widely different performance, which explains the wide range of deliverable capacities. Of course, optimal properties for O₂ will surely be very different than for H₂, CH₄ and CO₂. The possibility of visualizing over 1000 unique 5D plots interactively – using combined texted (which allows identifying the CSD code of any MOF), coloured and sized data points – allows delimiting the role of the “*combination of descriptors*” that influence not only oxygen storage but can be applied to any other gas. Compared to static plots typically found in the literature, this is the first time a web-domain data analysis tool is developed for screening MOFs.

The reviewer is correct when talking about the importance of charge density in gas adsorption. This is why we used this particular database of MOFs whose partial atomic charges are calculated from accurate ab initio derived electron density distributions (i.e. Density Derived Electrostatic and Chemical (DDEC) method). Since some of these MOFs present functional groups and therefore different surface chemistries, we are confident that the sheer number of structures probed in this report covers a large diversity, and that their effect is taken into account. Indeed, the combination of surface chemistry and textural properties affects oxygen uptake and the heat of adsorption.

We do not fully understand the comment about the trade-off between gravimetric and volumetric uptakes or a "hypothetical" study. With the ever-increasing emphasis on large-scale screening and computer-aided discovery of materials, we strongly believe our work provides significant insight and impact to the MOF community as a whole and specifically to those involved with computational screening. Note that we have screened ~3000 existing – not hypothetical – MOFs in this work. Most importantly, we did not limit our study to the computational screening, but moved to the synthesis and experimental test of the best material, confirming our prediction and approach. Therefore, the tools introduced here have substantial scientific merit as well as immediate relevance to today’s emphasis on big data analysis.

We have now added quantitative analysis of the effects of geometrical properties for the top materials in the manuscript; please see the response to Reviewer #1 and changes above.

The authors defined a lower limit for oxygen at 5 bar to determine the deliverable uptake which is clearly not understood. What is the constraint or the rational in doing so?

In practical applications of gas storage tanks, it is not possible to get all of the gas out of the tank (as this would require pulling a vacuum). There is always some lower pressure limit for the gas that can be delivered. This pressure is needed to "push" the gas exiting the tank through the device of interest. Thus, both a storage pressure and a delivery pressure are required to calculate the working capacity of an adsorbent. The delivery pressure could be as low as 1 bar, but with such a low pressure, one would expect the oxygen release kinetics to be very slow. By using 5 bar we aim to ensure the oxygen flow to be reasonable enough; a decrease in this lower limit of pressure would result in a higher working capacity for all the MOFs and in particular for our top one.

Another reason for using 5 bar is to allow us to compare our results with an earlier paper (Angew. Chem. Int. Ed., 53: 14092–14095), as described in our manuscript. However, given the reviewer's question, we have also performed oxygen adsorption simulations at 1 bar to explore further the low-pressure release regime and shed more light on the remaining unused oxygen in the tank for the ca. 3000 MOFs studied. Using the online MOF-explorer tool we developed, one can calculate how the deliverable capacity of e.g. UMCM-152 is increased from 250 cm³(STP)/cm³ to 269 cm³(STP)/cm³ when the release pressure is decreased from 5 bar to 1 bar and the storage pressure is kept fixed at 140 bar. We have now highlighted this point on page 11: **Using the online MOF explorer tool, one can also probe the effects of oxygen desorption pressure and monitor how the deliverable capacity of individual MOFs changes with respect to release pressure; for example, the deliverable capacity for UMCM-152 is increased from 250 cm³(STP)/cm³ to 270 cm³(STP)/cm³ when the release pressure is decreased from 5 bar to 1 bar and the storage pressure is kept fixed at 140 bar.**

Up to page 9 , there is a lot of claims about the “highest “, the “lowest” but there is no constructive and systematic scientific discussion about what is needed to make a better storage media for O₂.

We have included thorough investigation of textural properties that make MOFs to perform better. In addition, to include more in-depth analysis for the top materials we identified in this work, we further analysed the top 1 % of materials with optimum deliverable capacity as the storage pressure increases from 30 to 200 bar and further investigated the effects of MOFs textural properties on deliverable capacity; see Supplementary Figure 9, Figure 5 and Table S2 and relevant discussion above. We hope these additions will convince the reviewer about the importance of our work.

Reviewer #1 (Remarks to the Author):

I am reviewer 1.

I have reviewed the comments/feedback/modifications from P. Z. Moghadam et al., and appreciate the authors taking the time to address my concerns carefully. Overall, the revised text thoroughly addresses my concerns, and I strongly recommend this manuscript for publication in Nature Communications.

I have read reviewer 2 reports and the authors's responses.

I would like to offer that I believe the negative feedback of such referee to be largely unsubstantiated. It is clear that this person did not take the appropriate amount of time/effort to provide constructive feedback. My suggestion is to dismiss it.

Reviewer #2 (Remarks to the Author):

The present reviewer appreciate the great efforts deployed by the authors to address the reviewers' comments.

After a closer look at the paper and a painful exercise trying to associate the CCDC ref codes to real structures in order to make a chemical sense with the observed/predicated Oxygen capacity; the present reviewer comes to realize the need for additional corrections to the revised manuscript prior its final acceptance:

1- The present reviewer would appreciate to see some chemical analysis of the selected structures in the body of the manuscript where the composition of matter (organic and inorganic) is discussed and the pore geometry, size and shape are taken into account to understand and explain the observed and projected high Oxygen uptakes.

2- For the selected compound; please add a chemical name (MOF name) or a chemical formula next to the ref code as the ref code doesn't entail much to the reader without opening the SI or surfing the CCDC data base.

3- Most importantly and critical, I just come to realize that ref 18 has a compound with a very high gravimetric O₂ uptake and a higher deliverable capacity, higher than the ones in this study. Why this MOF doesn't appear in the current analysis? This should be the benchmark reference for the comparison and not reference 2 outdated to 2014. For example NU-125, the reference material that was referred to as a top candidate when stating a 22.5% increase for the presented UMCM-152 (ANUGIA), has a deliverable capacity of around 16 mol/Kg whereas the soc-MOF in ref.18 has a deliverable capacity over 20 mol/kg (i.e. 26 mol/kg much higher than 19.6 mol/kg for UMCM-152).

The present reviewer would like to see a clear explanation and incorporation of the comments above prior supporting the publication of the present manuscript.

Without a clear explanation and incorporation of the material in ref 18 in the comparison and analysis; the present reviewer doesn't see any scientific merit in publishing this study in its actual form.

Re: Decision on ID NCOMMS-17-21131A

Title of the paper: Computer-Aided Discovery of a Metal-Organic Framework with Superior Oxygen Uptake

We have examined and taken into consideration all of the comments returned to us by the reviewers regarding our paper. Each of the reviewer's comments is repeated below, in italics, along with our responses and the changes made to the manuscript, in red.

Reviewer #1 (Remarks to the Author):

I am reviewer 1.

I have reviewed the comments/feedback/modifications from P. Z. Moghadam et al., and appreciate the authors taking the time to address my concerns carefully. Overall, the revised text thoroughly addresses my concerns, and I strongly recommend this manuscript for publication in Nature Communications.

I have read reviewer 2 reports and the authors's responses.

I would like to offer that I believe the negative feedback of such referee to be largely unsubstantiated. It is clear that this person did not take the appropriate amount of time/effort to provide constructive feedback. My suggestion is to dismiss it.

We thank the reviewer once again for the positive feedback and the past excellent and constructive suggestions.

Reviewer #2 (Remarks to the Author):

The present reviewer appreciate the great efforts deployed by the authors to address the reviewers' comments.

After a closer look at the paper and a painful exercise trying to associate the CCDC ref codes to real structures in order to make a chemical sense with the observed/predicated Oxygen capacity; the present reviewer comes to realize the need for additional corrections to the revised manuscript prior its final acceptance:

1- The present reviewer would appreciate to see some chemical analysis of the selected structures in the body of the manuscript where the composition of matter (organic and inorganic) is discussed and the pore geometry, size and shape are taken into account to understand and explain the observed and projected high Oxygen uptakes.

A thorough investigation of textural properties that make MOFs perform better for oxygen storage is established in the manuscript. In the previous version we also included a more in-depth analysis for the top materials we identified in this work. We analysed the top 1% of materials with optimum deliverable capacity as the storage pressure increases from 30 to 200 bar and further investigated the effects of MOFs combination of textural descriptors (largest cavity diameter, surface area, void fraction, density)

on deliverable capacity – please, see discussion around Figure 5 in the manuscript. We believe this full discussion answers the current question.

2- For the selected compound; please add a chemical name (MOF name) or a chemical formula next to the ref code as the ref code doesn't entail much to the reader without opening the SI or surfing the CCDC data base.

This is an helpful suggestion. We have now linked all 2,932 structures studied in this work to the Cambridge Structural Database entries for easier structural access and analysis. Using the developed MOF explorer tool (available at: <http://aam.ceb.cam.ac.uk/mof-explorer>), users can click on each data point (i.e. MOF structure) to quickly access the MOF chemical formula and its original publication. We have highlighted this feature in the “data availability” section of the manuscript:

Page 17, “**Structure-property graphs can be viewed online at <http://aam.ceb.cam.ac.uk/mof-explorer>. All structures in the website are linked to the Cambridge Structural Database (CSD) for easier access and analysis.**”

3- Most importantly and critical, I just come to realize that ref 18 has a compound with a very high gravimetric O₂ uptake and a higher deliverable capacity, higher than the ones in this study. Why this MOF doesn't appear in the current analysis? This should be the benchmark reference for the comparison and not reference 2 outdated to 2014. For example NU-125, the reference material that was referred to as a top candidate when stating a 22.5% increase for the presented UMCM-152 (ANUGIA), has a deliverable capacity of around 16 mol/Kg whereas the soc-MOF in ref.18 has a deliverable capacity over 20 mol/kg (i.e. 26 mol/kg much higher than 19.6 mol/kg for UMCM-152).

The present reviewer would like to see a clear explanation and incorporation of the comments above prior supporting the publication of the present manuscript.

Without a clear explanation and incorporation of the material in ref 18 in the comparison and analysis; the present reviewer doesn't see any scientific merit in publishing this study in its actual form.

This is a very valid comment. We emphasise again that the 2,932 structures studied in this work are taken from a database of MOFs whose partial atomic charges are calculated from accurate and expensive ab initio derived electron density distributions (density derived electrostatic and chemical (DDEC) method; *Chem. Mater.* **2016**, 28, 785–793). Two considerations – and limitations – must be taken into account when screening this database. First, this database contains ca. 75% of the MOF materials reported in the original CoRE MOF database (*Chem. Mater.* **2014**, 26, 6185–6192). This is due to the fact that the charge density calculations for the large structures with primitive cells of several hundred atoms are computationally intensive and challenging; therefore, some of the large-unit cell and large-pore MOFs are not present in the DDEC database. Second, DDEC and CoRE MOF databases contain MOF materials published prior to 2014, and although it is not updated regularly, we are working with the Cambridge Crystallographic Database Centre to provide a solution to this issue (see

Chem. Mater. **2017**, 29, 2618–2625). Regarding the 25% of structures missing from DDEC database, although it is true that MOFs containing large pore volumes and surface areas show higher gravimetric adsorption capacities, their typical low densities will translate into very low volumetric capacities. For example, while the large porosity of the MOF identified by Reviewer #2, Al-soc-MOF-1 – not present in the DDEC MOF database – reported by Alezi and co-workers (*J. Am. Chem. Soc.* **2015**, 137, 13308–13318) presents a very high gravimetric deliverable capacity of 26.5 mol/kg of oxygen, its volumetric deliverable capacity is ca. 202 cm³(STP)/cm³, which is 22.8 % lower than that of UMCM-152, the top-performing structure identified in our work. As highlighted in the manuscript, we emphasise again that due to limited volume in storage tanks, the volumetric oxygen deliverable capacity – directly related to the size of the tank – is more crucial than gravimetric targets.

To clarify this point, we have added the above discussion to the manuscript and added the oxygen deliverable capacity values reported for Al-soc-MOF-1:

Page 6: “(...)We remark here that two considerations must be taken into account when screening the 2,932 structures present in the studied database. First, this database contains ca. 75% of the MOF materials reported in the original CoRE MOF database. This is due to the fact that the charge density calculations for the large structures with primitive cells of several hundred atoms are computationally intensive and challenging; therefore, some of the large-unit cell and large-pore MOFs are not present in this database. Second, DDEC and CoRE MOF databases contain MOF materials published prior to 2014. While MOFs containing large pore volumes and surface areas show higher gravimetric adsorption capacities, their typical low densities will translate into very low volumetric capacities.” and then in Page 8: “(...)Here, we emphasise again on the existence of a trade off between volumetric and gravimetric adsorption uptakes. For example while the large porosity of Al-soc-MOF-1,19 one of the top materials for oxygen storage found in the literature and not included in the DDEC database, presents a very high gravimetric deliverable capacity of 26.5 mol/kg of oxygen, its volumetric deliverable capacity is ca. 202 cm³(STP)/cm³, i.e. 22.8% lower than that of UMCM-152”.